# Curcumin Attenuates Lead-Induced Cerebellar Toxicity in Rats via Chelating Activity and Inhibition of Oxidative Stress

**DOI:** 10.3390/biom9090453

**Published:** 2019-09-06

**Authors:** Kabeer Abubakar, Maryam Muhammad Mailafiya, Abubakar Danmaigoro, Samaila Musa Chiroma, Ezamin Bin Abdul Rahim, Md Zuki Abu Bakar @ Zakaria

**Affiliations:** 1Department of Human Anatomy, Faculty of Medicine and Health Sciences, University Putra Malaysia, 43400 Serdang, Selangor Darul Ehsan, Malaysia; 2Department of Human Anatomy, College of Medical Sciences, Federal University Lafia, P.M.B 146 Akunza, Lafia, Nasarawa State, Nigeria; 3Department of Veterinary Anatomy, Faculty of Veterinary Medicine, Usman Danfodiyo University, P.M.B 2346 Sokoto, Nigeria; 4Department of Human Anatomy, Faculty of Basic Medical Sciences, University of Maiduguri, Borno State, Nigeria; 5Department of Radiology, Faculty of Medicine and Health Sciences, University Putra Malaysia, 43400 Serdang, Selangor Darul Ehsan, Malaysia; 6Department of Preclinical Sciences Faculty of Veterinary Medicine, University Putra Malaysia, 43400 Serdang, Selangor Darul Ehsan, Malaysia

**Keywords:** curcumin, lead toxicity, ICP-MS, horizontal bar, motor coordination, oxidative stress, cerebellum

## Abstract

Lead (Pb) is a toxic, environmental heavy metal that induces serious clinical defects in all organs, with the nervous system being its primary target. Curcumin is the main active constituent of turmeric rhizome (*Curcuma longa*) with strong antioxidant and anti-inflammatory properties. This study is aimed at evaluating the therapeutic potentials of curcumin on Pb-induced neurotoxicity. Thirty-six male Sprague Dawley rats were randomly assigned into five groups with 12 rats in the control (normal saline) and 6 rats in each of groups, i.e., the lead-treated group (LTG) (50 mg/kg lead acetate for four weeks), recovery group (RC) (50 mg/kg lead acetate for four weeks), treatment group 1 (Cur100) (50 mg/kg lead acetate for four weeks, followed by 100 mg/kg curcumin for four weeks) and treatment group 2 (Cur200) (50 mg/kg lead acetate for four weeks, followed by 200 mg/kg curcumin for four weeks). All experimental groups received oral treatment via orogastric tube on alternate days. Motor function was assessed using a horizontal bar method. The cerebellar concentration of Pb was evaluated using ICP-MS technique. Pb-administered rats showed a significant decrease in motor scores and Superoxide Dismutase (SOD) activity with increased Malondialdehyde (MDA) levels. In addition, a marked increase in cerebellar Pb concentration and alterations in the histological architecture of the cerebellar cortex layers were recorded. However, treatment with curcumin improved the motor score, reduced Pb concentration in the cerebellum, and ameliorated the markers of oxidative stress, as well as restored the histological architecture of the cerebellum. The results of this study suggest that curcumin attenuates Pb-induced neurotoxicity via inhibition of oxidative stress and chelating activity.

## 1. Introduction

Lead (Pb) is a ubiquitous environmental neurotoxin that induces several physiological, behavioral, and biochemical abnormalities in humans and animals [1]. Pb toxicity remains a common problem in both developing and industrialized countries due to unavoidable environmental and occupational exposure, resulting in about 600,000 new cases of intellectual disabilities in children and 143,000 deaths per year [1,2]. Although data from the Adults Blood Lead Epidemiology and Surveillance (ABLES) program indicate a significant decrease in the incidence of blood lead levels (BLLs) among adult industrial workers, occupational exposure to Pb remains a public health concern, accounting for about 94% of Pb exposure [3]. Occupational exposure to Pb is linked to several health consequences, such as cognitive impairment, reproductive disorders, hypertension, motor dysfunction, cancer, hepatotoxicity, nephrotoxicity, and mortality [4,5].

The mechanism of Pb toxicity is due to its ability to induce oxidative stress via disruption of the oxidant/antioxidant balance mechanism in cells [6]. Pb, a ubiquitous toxin, is known to induce oxidative stress by increasing the generation of reactive oxygen species (ROS), such as hydroxyl radicals, lipid peroxides, superoxide radicals, and hydrogen peroxide [6]. Pb toxicity has long been linked with impaired motor function, particularly deficits in visuomotor coordination among adult industrial workers and children exposed to Pb [7]. However, studies on the effect of Pb exposure on visuomotor integration among Yugoslavian and urban African American children showed that blood lead levels (BLLs) contributed significantly to poorer fine motor skills, as well as gross motor speed [7,8]. When treated with Pb, animal models display increased oxidative stress, cognitive impairments, degeneration of neurons, deficits in motor coordination, and mortality [9,10,11].

There has been great advancement in the field of phytotherapy, with marked increased in the use of medicinal plants such as curcumin as phytotherapeutics due to its pharmacological safety and antioxidant properties against heavy metal poisoning [12,13,14,15]. Chelation therapy is a recommended strategy in the treatment and management of chronic Pb poisoning, however, the major limiting factor regarding chelation therapy is some chelating agents could have a nonspecific action, resulting in the removal of essential metals from the body [16,17]. Further, application of chelation therapy in the treatment of chronic Pb poisoning is associated with several side effects, such as brain damage, anemia, liver and kidney diseases, anaphylactic shock, and others [16,18]. Curcumin is the main natural polyphenol in the rhizome of turmeric (*Curcuma longa*); it belongs to the family of ginger (*Zingiberaceae*), which is widely used as a traditional medicine and food in Asia [19]. It is a lipophilic, phenolic, and water-insoluble compound which has antioxidant, anti-inflammatory, and anti-cancer properties [20]. Curcumin exhibits its strong antioxidant properties by increasing the production of antioxidant enzymes, thus resulting in the scavenging of excess ROS and inhibition of lipid peroxidation [21]. Curcumin is lipophilic, thus it has the potential to cross the blood–brain barrier (BBB) and bind to plaques in the brain, thereby inhibiting amyloid-β peptide aggregation in Alzheimer’s disease [22]. Post-treatment with curcumin in subarachnoid hemorrhage (SAH)-induced mice preserved the integrity of the BBB and improved brain function via down-regulation of matrix metallopeptidase 9 (MMP-9) and inhibition of microglia cells, as well as reducing water content in the brain [23].

The mechanism of neuroprotection exerted by curcumin on neurodegenerative disorders is mainly due to its ability to bind redox-active transition metal ions, such as Mn^2+^, Fe^2+^, Cu^2+^, and Zn^2+^, to form active and tight complexes [24]. Additionally, curcumin has a wide range of pharmacological activities and a good safety margin and it is identified as a natural drug against neurodegenerative disorders [24]. However, there is a paucity of knowledge on the therapeutic potentials of curcumin in Pb-induced cerebellar toxicity. Therefore, the present study aimed at investigating the neurotherapeutic potential of curcumin on Pb-induced cerebellar toxicity in rats.

## 2. Materials and Methods

### 2.1. Chemicals

Lead acetate (CH_3_CO_2_)_2_Pb·3H_2_0, 99%) and animal feed were purchased from Sigma-Aldrich (St. Louis, MO, USA), while Malondialdehyde (MDA) ELISA and Superoxide Dismutase (SOD) assay kits were purchased from Elabscience (Houston, TX, USA). Curcumin, sodium acetate, aniline solution, and colophonium were purchased from Apical Scientific Sendirian. Berhad, Malaysia. All other chemicals used were of high analytical grade.

### 2.2. Animals

Thirty-six male Sprague Dawley rats aged 8 weeks, weighing 200–250 g, were obtained from the Animal Breeding Unit, Faculty of Veterinary Medicine, Universiti Putra Malaysia (UPM). The rats were kept in plastic cages and maintained at room temperature of 25 ± 2 °C with a 12 h light–dark cycle. All rats had free access to food and water during the study period. The rats were allowed to acclimatize for one week prior to the experiment in the animal house, Faculty of Medicine and Health Science, UPM. The animal management and handling procedures were performed based on the recommended institutional animal care and use committee (IACUC) guidelines with the reference number UPM/IACUC/AUP-R038/2018, approved on 19 September 2018.

### 2.3. Experimental Design

The rats were randomly divided into five groups (group A, B, C, D, and E) with six rats each in group B, C, D, and E, while group A consisted of twelve rats. Group A (normal control group), received normal saline orally for the whole of the experiment (8 weeks). Group B was designated as the lead-treated group (LTG) and received 50 mg/kg of Pb acetate orally for 4 weeks (induction of lead toxicity). Group C was designated as the recovery group (RC) and receives 50 mg/kg of Pb acetate orally for 4 weeks, followed by no treatment for 4 weeks. Group D, also known as treatment group 1 (Cur100) received 50 mg/kg of Pb acetate for 4 weeks orally, followed by 100 mg/kg of curcumin for another 4 weeks. Group E was designated as treatment group 2 (Cur200) and received 50 mg/kg of Pb acetate for 4 weeks orally, followed by 200 mg/kg of curcumin for another 4 weeks. At the end of the 4 weeks of oral administration of Pb acetate, six rats each from group A and group B were euthanized to confirm Pb toxicity via histopathological examination. At the end of the experiment (week 8) all rats were sacrificed and tissues were harvested for biochemical and histopathological examinations. Motor activity and weight of the rats was evaluated weekly (Figure 1). The dose of Pb acetate [25,26] and curcumin [27] was selected based on previous studies. The choice of oral administration of 50 mg/kg Pb acetate was chosen to mimic environmental exposure in humans; it is commonly used as Pb poisoning in rats models. Pb is present in different environmental media, such as air, water, sediments, and soil, although the major source of environmental exposure to Pb is attributed to drinking water [28]. However, the health consequences of Pb exposure depend on the cumulative dose and vulnerability of the individual rather than the environmental media that harbors the Pb [29]. Worthy to note is that populations that are highly susceptible to Pb exposure are affected even at low levels of exposure, which results in several pathologies, such as neurodegenerative disease, reproductive and hepato-renal disorders, decreased intelligence quotient (IQ) levels, and cardiovascular disease [28,30]. The doses of curcumin at 100 and 200 mg/kg in previous studies showed no sign of toxicity or morbidity, as they were effective in attenuation of nerve degeneration, motor dysfunction, cerebral ischemia, and impaired tight-junction protein integrity in rats [27,31,32]. Hence, curcumin 100 and 200 mg/kg were adopted in this study.

### 2.4. Motor Activity

Motor activity of the rats was evaluated using the horizontal bar method.

#### Horizontal Bar Method

The horizontal bar method is a test that measures forelimb strength and coordination in rodents. It involves the use of metal bars that were 2 mm in diameter and 38 cm long, suspended horizontally 49 cm above with support at two ends by a laboratory clamp and a padded surface to ensure soft landing of the experimental animals. The rats were placed on the center of the metal bar by handling them through their tail to ensure that only the forepaws grasped the metal bar. The falling time of the rats was recorded by a stopwatch and subsequently translated into scores as described by Deacon [33]. Scoring procedures used in the horizontal bar method are listed in Table 1.

### 2.5. Oxidative Stress Biomarker Analysis

#### 2.5.1. Determination of Protein Concentration

The total protein concentrations of the cerebellar homogenates were measured using the bicinchoninic acid assay (BCA assay). Bovine serum albumin (BSA) (2 mg/mL) was used as the standard with a working range of 125–2000 µg/mL

#### 2.5.2. Antioxidant Enzyme Activity Analysis

Superoxide dismutase (SOD) enzyme activity was determined in the rats’ cerebellum homogenates using an SOD assay kit (Elabscience, E-BC-K020). The reaction mixture consisted of 20 µL of tissue homogenates, 20 µL of enzyme working solution, and 200 µL of substrate application solution fully mixed and incubated at 37 °C for 20 min. The absorbance was measured at a wavelength of 450 nm using a micro-plate reader and the results were expressed as U/mg protein.

#### 2.5.3. Malondialdehyde (MDA) Analysis

The malondialdehyde level was determined from the rats’ cerebellums using an MDA ELISA kit (E-EL-006, Elabscience, Houston, TX, USA), using the principle of competitive ELISA. The test was conducted according to manufacturer’s manual (Elabscience, USA). Tissue pieces were washed, weighed, and homogenized in Phosphate-Buffered Saline (PBS) at a ratio of 1:9, with a glass homogenizer on ice. The homogenates were then centrifuged for 5 min at 5000× *g* to get the supernatant. The obtained supernatants were used to analyze the MDA using a micro-plate reader at a wavelength of 450 nm. The results for MDA were expressed as ng/mL.

### 2.6. Inductive Coupled Plasma Mass Spectrometry (ICP-MS)

#### 2.6.1. Sample Preparation

Harvested cerebellums from the rats were digested with 65% nitric acid using a microwave reaction system (Anton Paar, Multiwave PRO, Ashland, VA, USA),. About 0.5 g of the samples were placed in a Teflon vessel and 4 mL of nitric acid was added. The samples were then transferred into the microwave oven for 60 min to obtain a contamination-free, clear, digested solution. The solution was diluted with atomic water to 25 mL in agreement with digestion protocol as proposed by Rattanachongkiat et al. [34] and Simsek et al. [35].

#### 2.6.2. Sample analysis with ICP-MS

Pb was determined with an Agilent 7700× inductively-coupled plasma mass spectrometer (Agilent Technologies, Barcelona, Spain) equipped with a Micro Mist nebulizer (Glass Expansion, Melbourne, Australia). Table 2 shows the parameters and operation conditions of Agilent 7700× ICP-MS. The results were quantified using external calibration standards. Each sample was digested and analyzed in duplicate. Quality control (QC) was performed by analyzing from 20 part per billion (ppb) of calibration standard for every three samples.

### 2.7. Histopathological Examination and Scoring

The rats’ cerebellums were fixed in 10% buffered formalin for 5 days and subsequently prepared for histological examination. Paraffin sections of the cerebellar tissue were cut to 5 μm with a microtome (Leica 2235 Microtome, Buffalo Grove, IL. USA), mounted and stained on a glass slide with hematoxylin and eosin (H and E) or toluidine blue stain [36] for histochemistry examination under a light microscope (Leica DM4M, Brooklyn, NY, USA). Micrographs were captured using Moticam Pro 282A 5.0 MP (Motic images Software Plus 2.0 TWAIN, Hong Kong).

The numbers of necrotic Purkinje cells in the cerebellum were quantified using image analyzer software (Motic images Software Plus 2.0 TWAIN, Hong Kong). Quantification of the necrotic Purkinje cells was done at 400× magnification in ten non-overlapping fields from six different sections obtained from 3 rats from each group. The average values of the necrotic Purkinje cells for the 10 non-overlapping fields were calculated for each section and analyzed with GraphPad Prism. Counting of the necrotic Purkinje cells was done manually with the aid of an image analyzer in a blinded manner by an independent pathologist.

### 2.8. Statistical Analysis

The data obtained from this study were analyzed using unpaired independent student’s t-test and one-way and two-way ANOVA using GraphPad Prism (GraphPad Prism software, Inc. Version 6.01, San Diego, California, USA). Data were presented as the mean ± standard error of the mean (SEM), where *p* < 0.05 was considered statistically significant.

## 3. Results

### 3.1. Induction of Pb Acetate Toxicity in Rats

#### 3.1.1. Effect of Pb Acetate on Body Weight of Rats during Pb Toxicity Induction

In order to evaluate the effect of Pb acetate on body weight of the experimental rats, two-way ANOVA was employed and the results indicated a statistically significant interaction between the effect of Pb acetate on body weight and weeks of induction of Pb toxicity, (F (16, 125) = 3.01, *p* = 0.0003). Tukey’s post hoc comparison indicated a significant decrease (*p* < 0.05) in body weight in group B, C, D, and E rats in week 3 and week 4 when compared to the body weight of rats in group A, as shown in Figure 2.

#### 3.1.2. Effect of Pb Acetate on Motor Score and Coordination in Rats during Pb Toxicity Induction

In order to evaluate the effect of oral administration of Pb acetate on the motor function of rats, the horizontal bar test was performed. The results indicated a statistically significant interaction between treatment effect of oral administration of Pb acetate and weeks of treatment (F (16, 125) = 2.23, *p* = 0.0072) in the motor scores among the rat groups. Tukey’s post hoc comparison showed a significant decrease (*p* < 0.05) in the motor score of Pb-administered rats regarding their ability to maintain a hand grip balance on the 2 mm horizontal bar at week 3 and week 4 in group B, C, D, and E rats compared with the motor score of rats in group A, as seen in Figure 3.

#### 3.1.3. Effect of Pb Acetate on Oxidative Stress Status of Cerebellum in Rats of Control and LTGs during Pb Toxicity Induction

To confirm the induction of Pb toxicity in the rats, the oxidative stress status in their cerebellums were assessed through unpaired independent student’s t-test.

##### SOD Activity

Results from the unpaired independent student’s t-test (Figure 4**)** showed a statistically significant difference in cerebellar SOD activity between the rats of the control group (*M* = 8.4, SEM = 0.4) and the LTG (*M* = 4.833, SEM = 0.3283); t (4) = 6.892, *p* = 0.0023 following oral administration of Pb acetate.

##### MDA Level

Similarly, the results of the unpaired independent student’s t-test results, shown in Figure 5, indicated a statistically significant difference in cerebellar MDA levels of rats in the control group (*M* = 15.4, SEM = 2.794) and the LTG (*M* = 42.19, SEM = 7.979); t (4) = 3.169, *p* = 0.0339 after oral administration of lead acetate for four weeks.

#### 3.1.4. Determination of Pb Concentration in the Cerebellums of Rats in the Control Group and LTG after Pb-Toxicity Induction

To determine the concentration of Pb in the rats’ cerebellums, the ICP-MS technique was used. Independent student’s t-test was employed to analyze the difference in Pb concentration in the cerebellums of rats in the control group and LTG after oral administration of Pb acetate. The results indicated statistically significant differences in Pb concentrations in the cerebellums of rats in the control group (*M* = 0.2, SEM = 0.024) and LTG (*M* = 2.58, SEM = 0.6009); t (4) = 3.981, *p* = 0.0164 (Figure 6).

#### 3.1.5. Effect of Pb Acetate on Histology of Cerebellum of Rats in the Control Group and LTG after the Induction of Pb Toxicity

##### Cerebellum Stained with Hematoxylin and Eosin (H and E)

The histopathological examination of the cerebellar tissue of Pb-administered rats using H and E stain revealed histological alterations of the cerebellar cortex layers with shrinkage and degeneration of the Purkinje and molecular layer cells with scattered glial cells. The Purkinje cells exhibited darkly stained nuclei with eosinophilic cytoplasm and empty spaces between them, indicating degeneration of the Purkinje cells (Figure 7B) compared with the control rats (Figure 7A).

Further, in order to evaluate the effect of Pb acetate on the Purkinje cells of the cerebellum, the nonparametric t-test was used. The Mann–Whitney test indicated that the number of degenerated Purkinje cells was significantly greater in the cerebellum of rats in the LTG (Mdn = 5.5) compared to the control group (Mdn = 0.73); U = 0, *p* = 0.0022 (Figure 7C).

##### Cerebellum Stained with Toluidine Blue

In order to further confirm the neurodegenerative effect of Pb acetate induction on the cerebellums of rats, toluidine blue staining was also performed. The histological results from the cerebellums of rats in the LTG indicated degeneration of the cells of the molecular layer and distortion of the Purkinje cell layer, with the Purkinje cells having a darkly stained cytoplasm and distorted nuclei with empty space between them, indicating loss of Purkinje cells (Figure 8B) compared with rats in the control group (Figure 8A).

##### Effect of Pb Acetate on the Weight of the Cerebellums of Control Group and LTG Rats after Pb-Toxicity Induction

An unpaired t-test was employed to evaluate the effect of Pb acetate administration on cerebellar weight in rats of the control group and LTG. The results showed statistically significant differences in the cerebellar weights of rats in the control group (*M* = 0.5333, SEM = 0.02418) when compared to LTG rat (*M* = 0.4033, SEM = 0.1706); t (10) = 4.393, *p* = 0.0013 (Figure 9).

### 3.2. Treatments of Pb Acetate-Induced Rats with Curcumin

#### 3.2.1. Effect of Curcumin on the Body Weight of Pb-Induced Rats

Two-way ANOVA results showed a statistically significant interaction between the effect of curcumin on the body weight of rats and the weeks of administration (F (24,180) = 3.242, *p* = 0.0001). Tukey’s post hoc test indicated a significant decrease (*p* < 0.05) in the body weight of rats in the RC and Cur100 and Cur200 groups in week 3, week 4, week 5, week 7, and week 8 when compared to the body weight of rats in the control group, as shown in Figure 10.

#### 3.2.2. Curcumin Ameliorates Pb-Induced Alteration of Motor Coordination of Rats in the Horizontal Bar Test

The two-way ANOVA results showed a statistically significant interaction between the effect of oral administration of curcumin and weeks of treatment (F (24, 180) = 2.448, *p* = 0.0004) in the motor scores of Pb-induced rats treated with curcumin (Figure 11). Tukey’s post hoc comparison indicated a statistically significant decrease (*p* < 0.05) in the motor score of rats in the RC, Cur100, and Cur200 groups regarding their ability to maintain a hand grip balance on the 2 mm horizontal bar in week 3, week 4, week 5, and week 6 when compared to the control group of rats. Additionally, Tukey’s post hoc test further revealed a significant increase (*p* < 0.05) in the motor score of rats in the control group in week 7 and week 8 compared to the motor score of rats in the RC group.

#### 3.2.3. Curcumin Reverse Pb-Induced Oxidative Stress in Rats’ Cerebellums

In order to assess the antioxidant properties of curcumin on Pb-induced oxidative stress in rats, the cerebellar homogenates of the rats were analyzed for SOD activity and MDA levels.

##### SOD Activity

As shown in Figure 12, the one-way ANOVA results revealed a statistically significant differences in SOD activity in the cerebellums of different experimental rats groups (F (3, 8) = 3.768, *p* = 0.0493). Tukey’s post hoc test showed a statistically significant decrease in SOD activity in the cerebellums of the RC rats (5.167 ± 0.133, *p* = 0.0443) when compared with the control group of rats (8.4 ± 0.4).

##### MDA Level

The one-way ANOVA result showed a statistically significant difference in MDA levels in the cerebellums of all rat groups (F (3, 8) = 5.844, *p* = 0.0205). Tukey’s comparison test revealed a significant increase in the MDA levels in the cerebellums of rats from the RC group (22.78 ± 1.579, *p* = 0.0153) compared to the control group of rats (15.4 ± 1.062), as shown in Figure 13.

#### 3.2.4. Chelating Potentials of Curcumin on Pb-Induced Toxicity in Rats

The results obtained from the ICP-MS analysis were subjected to one-way ANOVA to evaluate the mean Pb concentration in the cerebellums of rats in the control, RC, Cur100, and Cur200 groups. Results obtained showed statistically significant differences in Pb concentrations in the cerebellums of the different rat groups (F (3, 8) = 8.61, *p* = 0.0069). Tukey’s post hoc comparison indicated significant decreases in Pb concentration in the cerebellums of rats in the control (0.1828 ± 0.02414, *p* = 0.0001), Cur100 (0.6319 ± 0.1545, *p* = 0.0014), and Cur200 (0.4848 ± 0.08147, *p* = 0.006) groups compared to rats in the RC group (Figure 14).

#### 3.2.5. Curcumin Attenuates Pb-Induced Cerebellar Damage in Rat’s Cerebellum

##### H and E Staining

The histology results from the cerebellums of rats in the RC group stained with H and E after withdrawal of Pb acetate indicated no recovery when compared to rats in the control group (Figure 15A). The RC group cerebellums revealed eosinophilic Purkinje cells with dark and irregular nuclei, with the molecular layer appearing to have scattered glial cells with perineural spaces; the granular layer appeared to have a normal histological appearance (Figure 15B).

Oral administration of curcumin for four weeks attenuated the pathological changes in the cerebellums of rats in the Cur100 (Figure 15C) and Cur200 (Figure 15D) groups. In addition, semi-quantitative analysis of the Purkinje cells of the cerebellums revealed statistically significant differences in degenerated Purkinje cells of the experimental rats (H (3) = 19.75, *p* = 0.0002) with a mean rank of 3.583 for the control rats, 21.5 for RC rats, 13.83 for Cur100 rats, and 11.08 for Cur200 rats. Dunn’s multiple comparison test further showed statistically significant increases (4.333 ± 0.2246, *p* = 0.0001) in degenerated Purkinje cells in the cerebellums of RC rats compared to the rats in the control group (Figure 15E).

##### Toluidine Blue Staining

The photomicrograph sections from the cerebellums of the control group of rats indicated normal histological structure of the cerebellum with the three layers of the cerebellar cortex. The Purkinje cells appeared to have regular and prominent central nuclei. The granular and molecular layers showed normal cells with darkly stained nuclei (Figure 16A). Photomicrograph sections from the cerebellums of the RC rats revealed alterations in the Purkinje cell layer, with the Purkinje cells appearing to have darkly stained cytoplasm with irregular nuclei. The molecular layer showed scattered glial cells, although the granular layer appeared to be normal (Figure 16B).

The photomicrograph section from the Cur100 and Cur200 showed restoration of the Purkinje cell layer with healthy Purkinje cells. The molecular and granular layers appeared to be normal with normal cells (Figure 16C,D).

##### Effects of Curcumin Administration on Cerebellar Weight

One-way ANOVA revealed statistically significant differences in cerebellar weights between the groups of rats (F (3. 20) = 3.195, *p* = 0.457). Tukey’s post hoc comparison revealed a statistically significant decrease in cerebellar weight of the RC group of rats (0.4617 ± 0.01778, *p* = 0.0486) compared to the control group, as shown in Figure 17.

## 4. Discussion

Preceding studies have documented the toxic effects of Pb on biological systems resulting in several pathologies and clinical implications with morbidity on almost all organs, with the brain, kidney, and liver serving as primary targets [18,37,38]. These pathological alterations may include increased oxidative stress [9], neurological deficits [39], decreased motor coordination, and cognitive deficits [40,41]. Alterations in astrocyte maturation, degeneration of neural cells [39,41], renal dysfunction, degenerative changes in tubular epithelium, and hepatotoxicity [2,42] were also reported in Pb poisoning, both in humans and different rodent species. In the present study, a rat model was used to investigate the therapeutic potentials of curcumin on the fundamental mechanisms of motor dysfunction and neurodegeneration caused by Pb toxicity. The Pb-induced rats developed a remarkable deficiency in the cerebellum-dependent horizontal bar test for motor function, increased oxidative stress, marked degeneration of cells in molecular and Purkinje cell layers of the cerebellum, and high concentrations of Pb in their cerebellums. However, treatment with curcumin, irrespective of the dose given, attenuated the aforementioned alterations and aberrations caused by Pb acetate-induced toxicity in the rats.

Worthy to note in this study is the choice of male Sprague Dawley rats for all the experimentations. As female sex hormones, such as estrogen, prolactin, and progesterone, may influence cognitive function, emotion, and motor behavior in rats because their cyclical hormonal changes usually come with mood swings. Hence, female rats were less preferred by the authors for pharmacological testing of phytochemical substances like curcumin due to the potential for confounding factors [43,44,45]. Therefore, healthy male Sprague Dawley rats were used in this study to evaluate the neurotherapeutic effects of oral administration of curcumin on Pb-induced toxicity of motor function and coordination.

Coordination of motor behavior and cognitive function depend on the integrity of the nervous system and core psychological functions of the animals [33,46]. The cerebellum is a delicate structure that is vulnerable to intoxication and poisoning. The Purkinje cells are particularly susceptible to injury after exposure to environmental toxins such as Pb [47]. Further, the cerebellum plays a vital role in the unification of motor and sensory functions; any lesion on the cerebellum may result in deterioration of motor coordination and balance [48]. In this study, a decrease in motor score on the horizontal bar test was observed in Pb-induced rats, which is in agreement with the previous works documented by Nehru and Sidhu [49], Barkur and Bairy [50], and Sabbar et al. [51], who also reported a decrease in cognitive and motor functions in rat models regarding Pb toxicity. Mason et al. [7] also reported that Pb exposure affected motor function, such as the deficits seen in visuomotor coordination in adult Pb workers and children exposed to Pb. The mechanism in which Pb executes this decrease in motor function could be due to its ability to induce oxidative stress, as seen in rats exposed to Pb [6]. However, the present study observed an increase in the motor score of rats treated with curcumin, regardless of the dose given. Moore et al. [52], also observed similar findings and attributed it to the antioxidant and anti-inflammatory properties of curcumin. Chongtham and Agrawal [53] also reported that curcumin ameliorated disease symptoms in a *Drosophila* model of Huntington disease (HD). However, supplementation of diet with curcumin for 12 weeks in a healthy aging population did not influence motor performance [54].

Alterations of oxidative status, either by overproduction of oxidants or deficits in antioxidant activity, could be one of the direct consequences of Pb toxicity and poisoning in living organisms [6]. Preceding studies indicate that oxidative stress is linked to the pathogenesis of Pb toxicity, resulting in lipid peroxidation [6], neurodegeneration [55], oxidation of hemoglobin [56], and impairment of fundamental biological cellular processes, such as cell adhesion, intra- and inter cellular signaling, ionic transportation, enzyme regulation, neurotransmitter release, and apoptosis [18]. The fundamental oxidants that play a vital role in redox reactions include hydroxyl radicals (OH), hydroxyl anions (OH^−^), hydrogen peroxide (H_2_O_2_), nitric oxide (NO), and peroxynitrite (ONOO^−^) [57]. MDA or thiobarbituric acid-reactive species (TBARS) are the end products of lipid peroxidation that play a vital role in lipid membrane damage in cells due to increased reactive oxygen species (ROS) [6]. The present study revealed an increased level of MDA and decreased SOD activity in the cerebellum of Pb-induced rats. However, these alterations were ameliorated by treatment with curcumin, irrespective of the dose given. These findings were in agreement with previous studies that have reported the antioxidant properties of curcumin [58,59,60].

Inductive-coupled plasma mass spectrometry (ICP-MS) is a robust technique for the molecular analysis of elements and physiochemical compounds, using separation techniques that make it suitable for detecting elements in pharmaceutical research and for specific investigations of elements present in molecules [61]. ICP-MS is a multi-element system technique in the analysis of biological fluids with greater sensitivity and selectivity when compared to inductive-coupled plasma-optical emission spectrometry (ICP-OES) and graphite furnace atomic absorption spectrometry (GF-AAS) [35]. Concentration of trace elements beyond physiological limits in organs can be toxic in both animals and humans. Likewise, concentration of heavy metals such as Pb in the biological system are known to be toxic and affect biochemical reactions [35]. Further, bones remain a vital site for Pb accumulation after exposure, although circulating Pb in the blood can be distributed to various vital organs such as the brain, kidneys, and liver [62,63]. This present study decided to use ICP-MS for its superiority over other techniques and found significantly higher concentrations of Pb in the cerebellums of Pb-induced rats when compared to their control counterparts. These results are in agreement with the previous works of Flora et al. [16], Sousa et al. [63], and Simsek et al. [35], who reported increased concentrations of Pb in the brains of Pb-induced rats. A noteworthy finding was the administration of curcumin to Pb-induced rats, which decreased the concentrations of Pb in their cerebellums. These findings were in accordance with the previous studies of Daniel et al. [64], Flora et al. [16], Mary et al. [24], and Shen et al. [65], who reported the chelating properties of curcumin. Daniel et al. [64] provided a background understanding of the chelating properties of curcumin on Pb, resulting in a reduction of Pb concentration in the brain of rats by forming metal–ligand complexes through either directly bonding to the Pb molecules or via intermolecular hydrogen bonding. Further, the results obtained by Mary et al. [24] revealed that curcumin formed active chelates with zinc ions and suggested that curcumin as a multipotent agent could be best used in chelation therapies for numerous neurodegenerative diseases, such as Alzheimer’s disease, and a radical scavenger where zinc ions are found in abundance. In addition, Shen et al. [65] documented the chelating properties of curcumin on copper (II) (Cu(II)) metal ions, which are implicated in the pathogenesis of Alzheimer’s disease. Further, Cu(II) binds to at least two curcumin molecules [66]; the enolic proton in curcumin is readily dissociated in solution [67] as well as inspired by the 1:2 manganese (II) (Mn(II)) chelating mode of curcumin [68]. Interestingly, curcumin chelating potentials were also documented in other tissues, such as the liver, kidneys, and blood [16]. Flora et al. [16] assessed the protective effect of curcumin and nanocurcumin on Pb-induced toxicity in the kidneys, livers, blood, and brains of mice. Further, the administration of curcumin and nanocurcumin revealed beneficial effects by attenuating the pathological lesion caused by toxic effect of Pb, as well as reducing the concentration of Pb in the liver, brain, kidneys, and blood, although nanocurcumin showed more of a chelating effect than curcumin. Hence, the protective effect of curcumin was chiefly attributed to its scavenging of free radicals and chelating properties [15]. In addition, co-treatment with curcumin in female Kunming mice exposed to sodium arsenite in drinking water revealed amelioration of the adverse effect of arsenic toxicity on the liver and accelerated the excretion of arsenic via the urinary system [69,70]. Agarwal et al. [71] demonstrated in their work that pre-treatment and post-treatment with curcumin in rat models protected their livers and kidneys against mercury-induced toxicity, hence, curcumin chelated the mercury, resulting in a decreased concentration of mercury in the liver and kidneys.

In the present study, multiple pathological lesions were observed in the cerebellums of Pb-induced rats. These changes included neuronal damage and alterations of the histological architecture of the cerebellar cortex. The Purkinje cells exhibited eosinophilic cytoplasm with darkly stained and irregular nuclei, leaving empty spaces between them. Semi-quantitative analysis of the Purkinje cells showed a significant increase (*p* < 0.05) in the number of necrotic Purkinje cells. Further, Pb-induced rats showed decreased (*p* < 0.05) cerebellar weights, which could be attributed to the marked degeneration of the cerebellar cells observed. These results are in accordance with several preceding studies [9,37,48,72,73] that reported similar findings of neuronal degeneration in Pb and other heavy metal toxicity on the cerebellums, with more consequences for the Purkinje cells due to sensitivity of the Purkinje cell layer. Further, previous studies also documented the toxic effect of Pb administration on juvenile and young rats, resulting in different neurological impairments in various parts of the brain and nervous system [74,75,76,77]. However, treatment with curcumin reversed the above-mentioned Pb-induced morphological aberrations in the rats. The results obtained from this study are in accordance with the results documented in previous studies on the attenuating effects of curcumin on Pb-induced neurotoxicity [64,77,78,79]. These suggest that curcumin has antioxidant properties, as seen in the present study, or a combined antioxidant and anti-inflammatory role in neurotoxicity and neurodegenerative diseases, as documented by previous studies [22,80,81].

The most important step in dealing with Pb toxicity is to avoid exposure to the sources of Pb contamination, which might not always be feasible. Therefore, standard drug chelators on the market exist, which are being used for heavy metal poisoning; however, they are expensive and exhibit many adverse effects [17,18]. This study revealed that withdrawal of Pb acetate alone was not enough to restore the damage inflicted by Pb toxicity on rats of the RC group. The persistent changes observed after Pb withdrawal include morphological aberrations of the cerebellum, decrease in motor scores on the horizontal bar test, oxidative stress, and high concentrations of Pb in the cerebellum as observed by ICP-MS analysis compared to the curcumin-treated groups. These could be due to specific kinetics, where a proportion of the absorbed Pb accumulates in the bone [11,30,82] and is gradually released to soft tissues, such as the brain, kidneys, and liver [30,82,83]. Similar findings were reported by Omobowale [84], Nehru and Sidhu [49], and Khalaf et al. [85] after withdrawal of Pb exposure in rat models of Pb toxicity.

## 5. Conclusions

In consideration of the results obtained from this study, Pb toxicity resulted in decreased motor function, increased oxidative stress, aberrations in the histological structure of the cerebellum, and accumulation of Pb in the cerebellums of affected rats. Treatment with curcumin, regardless of the dose, attenuated the abnormalities caused by Pb toxicity, which could be due to the antioxidant and chelating properties of curcumin (Figure 18). To conclude, curcumin could be developed as a natural drug for treatment of Pb toxicity due to its therapeutic potential and wide pharmacological safety margin. Finally, the authors recommend future research on the activation of astrocytes and microglia involved in neurodegenerative diseases, as well as longer treatment periods with curcumin to assess the effectiveness of its long-term preclinical applications in Pb-induced toxicity and neurodegeneration.

## Figures and Tables

**Figure 1 biomolecules-09-00453-f001:**
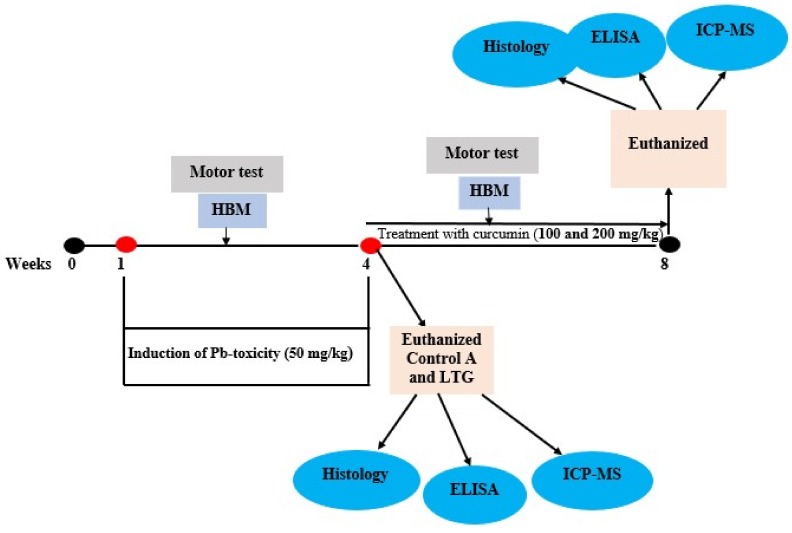
Schematic representation of the experimental design. HBM: Horizontal bar method; LTG: Lead-treated group; ICP-MS: Inductive coupled plasma spectrometry.

**Figure 2 biomolecules-09-00453-f002:**
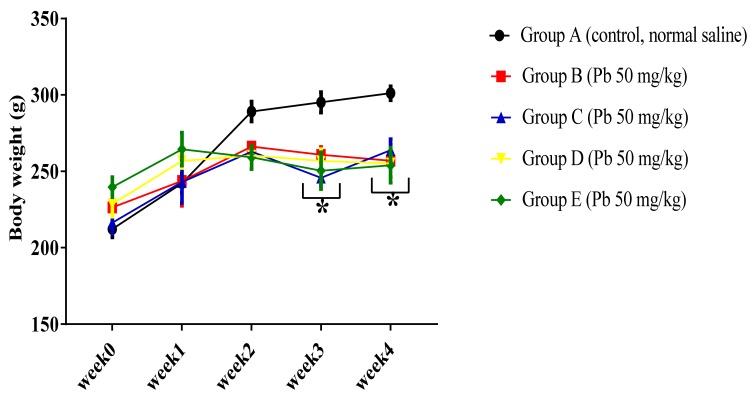
Effect of Pb toxicity on body weight following four weeks of oral administration Pb acetate. Data are represented as mean ± SEM (*n* = 6); * *p* < 0.05 vs. control.

**Figure 3 biomolecules-09-00453-f003:**
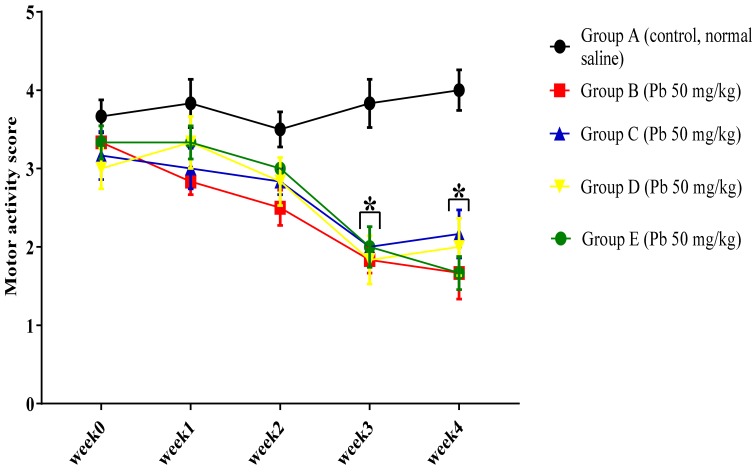
Effect of oral administration of Pb acetate on motor score of rats during induction of Pb toxicity. Values are presented as mean ± SEM (*n* = 6); * *p* < 0.05 vs. control.

**Figure 4 biomolecules-09-00453-f004:**
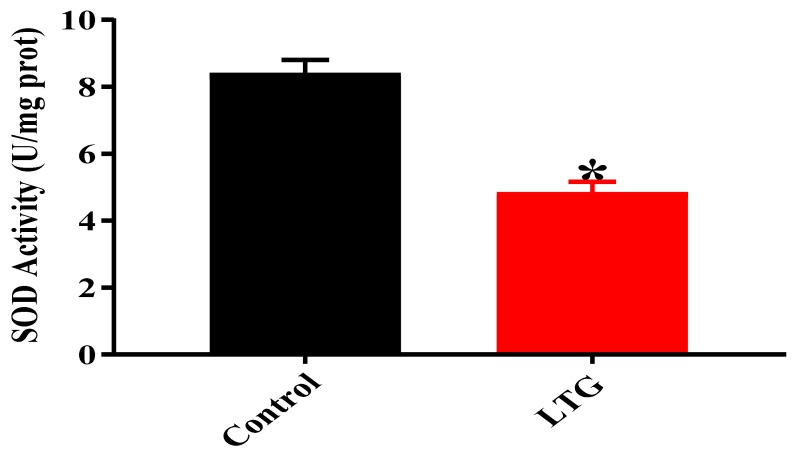
Effect of Pb acetate administration on SOD activity in the cerebellums of rats after Pb toxicity induction. Data are expressed as mean ± SEM (*n* = 3); * *p* < 0.005 vs. control.

**Figure 5 biomolecules-09-00453-f005:**
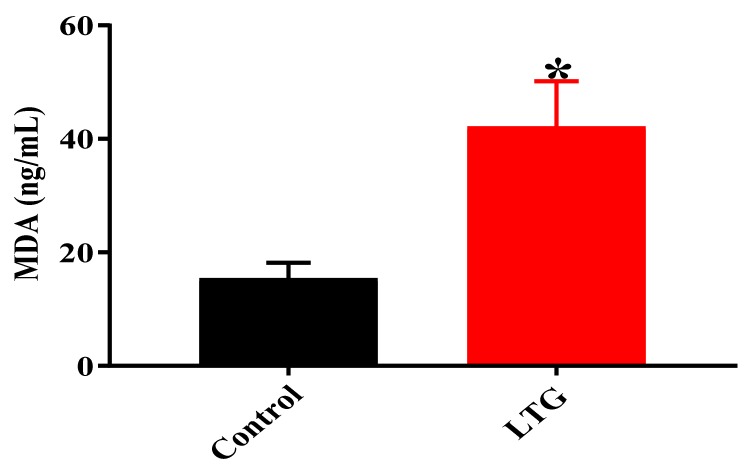
Effect of Pb acetate administration on MDA levels in the cerebellums of rats after Pb toxicity induction. Data are expressed as mean ± SEM (*n* = 3); * *p* < 0.005 vs. control.

**Figure 6 biomolecules-09-00453-f006:**
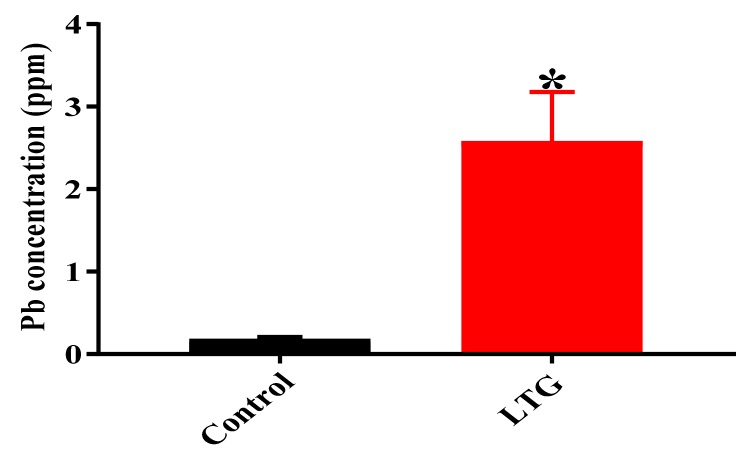
Concentration (ppm) of Pb in rats’ cerebellums after induction of Pb toxicity for four weeks. Data are represented as mean ± SEM (*n* = 3); * *p* < 0.05 vs. control.

**Figure 7 biomolecules-09-00453-f007:**
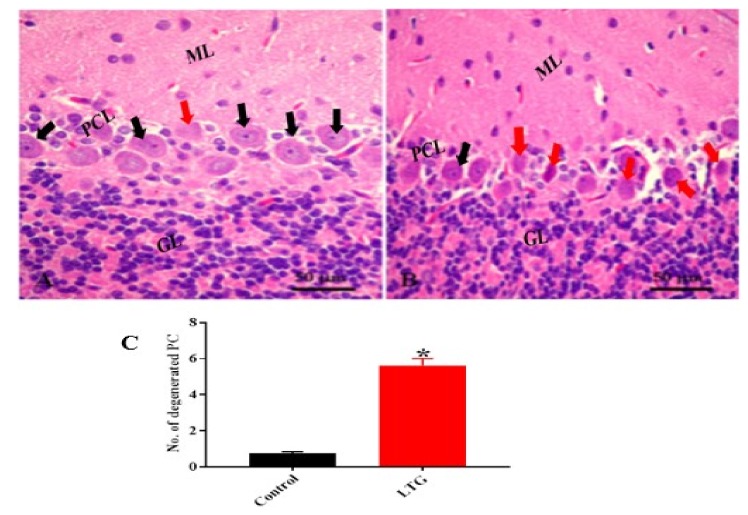
(**A**) Photomicrograph of control cerebellar cortex showing the layers of the cerebellum: The molecular layer (ML), the middle Purkinje cell layer (PCL) with a large pyriform shape (black arrow), and the inner granular layer (GL) with aggregation of granular cells. (**B**) Lead-treated group (LTG) showing marked degeneration of Purkinje cells (red arrow). (**C**) Semi-quantitative representation of degenerated Purkinje cells of the control group and LTG. Data are represented as mean ± SEM (*n* = 3); * *p* < 0.05 vs. control. H and E 400×, scale bar = 50 µm.

**Figure 8 biomolecules-09-00453-f008:**
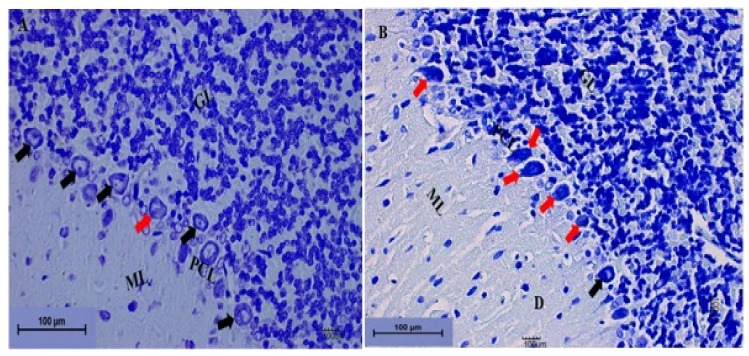
Photomicrograph sections of the cerebellums from the rat groups. (**A**) Control showing the molecular layer (ML), the granular layer (GL), and the Purkinje cell layer (PCL). The Purkinje cells are shown by the black arrow, with prominent nuclei. (**B**) LTG showing the molecular layer (ML) with a degenerated area of cells (D) and Purkinje cells with darkly stained cytoplasm and distorted nuclei leaving spaces between them (red arrow). The granular layer (GL) indicates deeply stained cells with vacuolation (toluidine blue 200×, scale bar = 100 µm).

**Figure 9 biomolecules-09-00453-f009:**
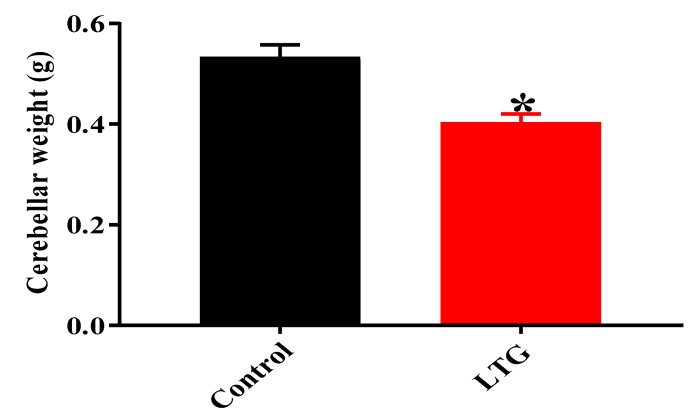
Effect of Pb acetate on cerebellar weights of rats of the control group and LTG. Data are represented as mean ± SEM (*n* = 6); * *p* < 0.05 vs. control.

**Figure 10 biomolecules-09-00453-f010:**
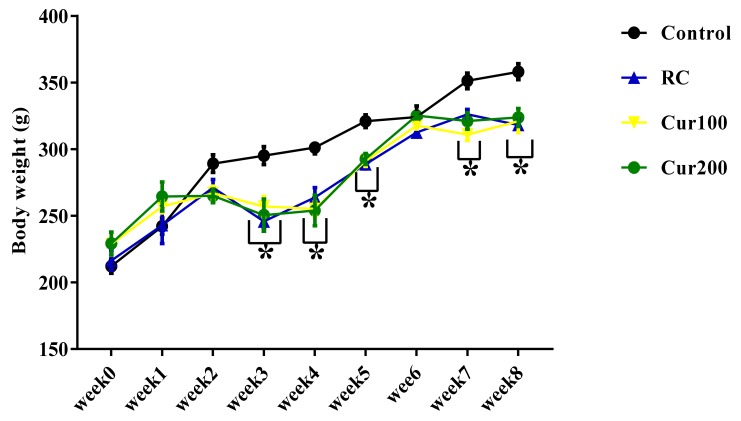
Effect of oral administration of curcumin on body weight of Pb acetate-induced rats. Data are represented as mean ± SEM (*n* = 6); * *p* < 0.05 vs. control.

**Figure 11 biomolecules-09-00453-f011:**
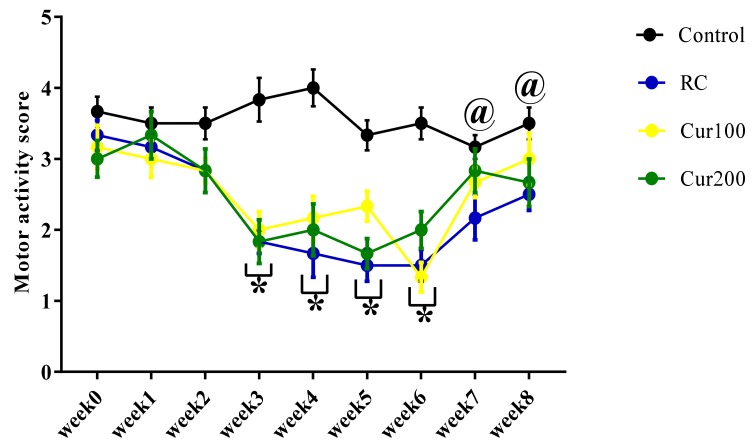
Effect of oral admiration of curcumin on Pb-induced rats in the horizontal bar test. Values are presented as mean ± SEM (*n* = 6); * *p* < 0.05 vs. control, ^@^
*p* < 0.05 vs. RC.

**Figure 12 biomolecules-09-00453-f012:**
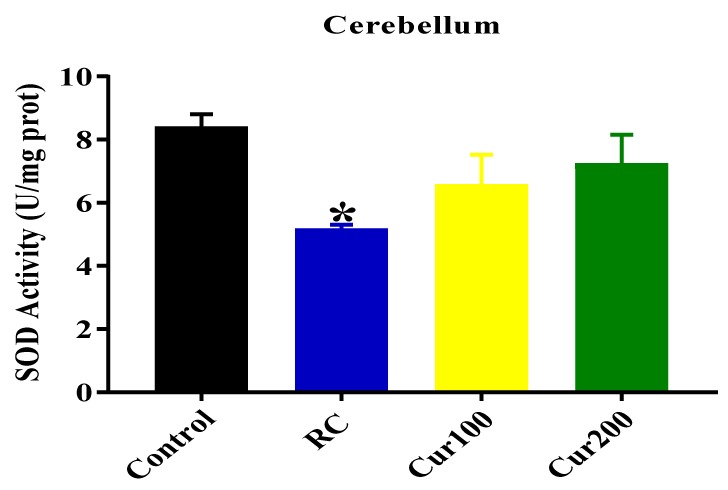
Effect of curcumin on superoxide dismutase (SOD) activity in the cerebellums of Pb-induced rats. Data are expressed as mean ± SEM (*n* = 3); * *p* < 0.05 vs. control.

**Figure 13 biomolecules-09-00453-f013:**
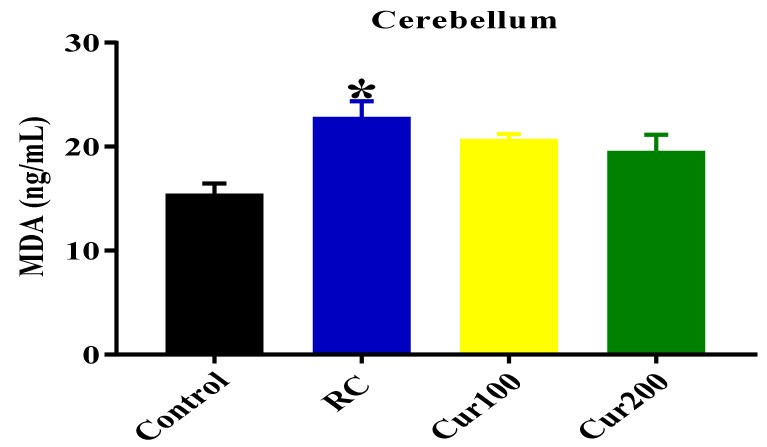
Effect of curcumin on malondialdehyde (MDA) levels in the cerebellums of Pb-induced rats. Data are expressed as mean ± SEM (*n* = 3); * *p* < 0.05 vs. control.

**Figure 14 biomolecules-09-00453-f014:**
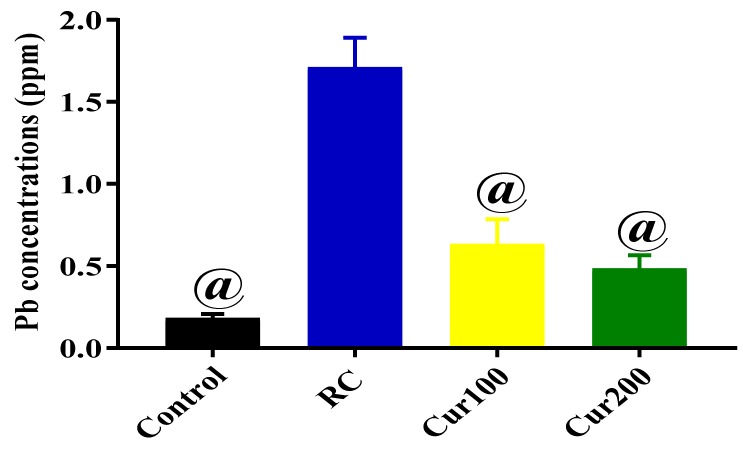
Concentration of Pb in rats’ cerebellums after oral administration of curcumin and withdrawal of Pb acetate for four weeks. Data are represented as mean ± SEM (n = 3); ^@^
*p* < 0.05 vs. RC.

**Figure 15 biomolecules-09-00453-f015:**
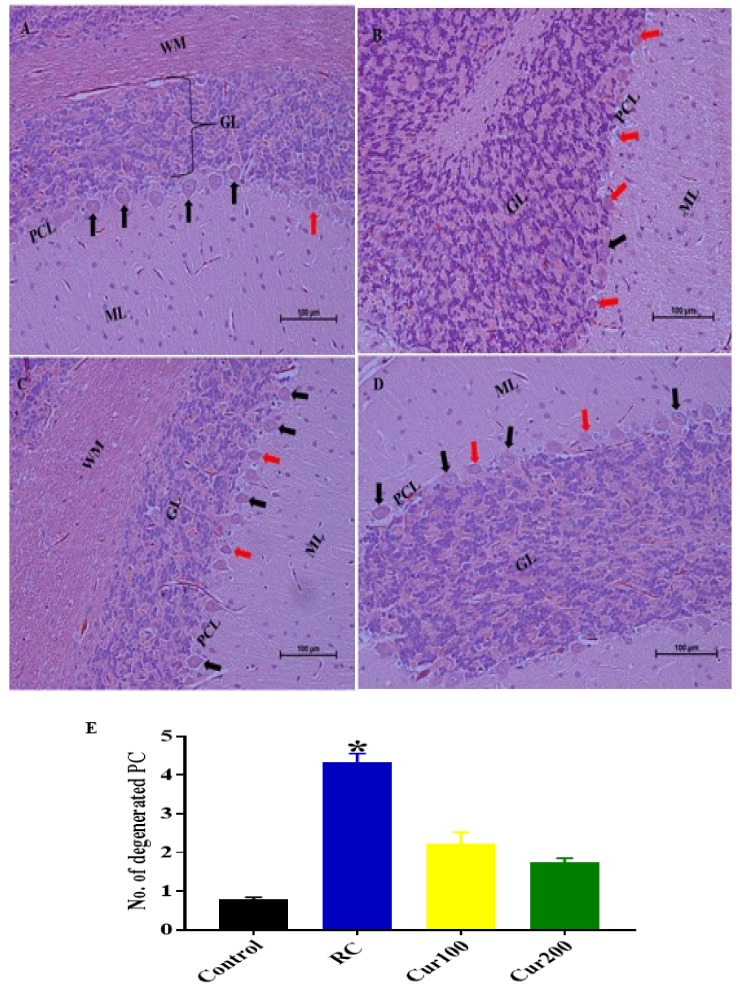
Photomicrograph sections of the cerebellums of the rat groups. (**A**) Control group indicating layers of the cerebellar cortex, the molecular layer (ML) with glial cells, the middle Purkinje cell layer (PCL) with the Purkinje cells having a large pyriform shape (black arrow), white matter (WM), and the inner granular layer with aggregation of granular cells (GL). (**B**) RC showing eosinophilic Purkinje cells with irregular dark cytoplasm (red arrow) and scattered glial cells in the molecular layer (ML). (**C**) Cur100 showing the Purkinje cells with prominent nuclei and a regular shape (black arrow); the molecular and granular layers appear normal. (**D**) Cur200 showing normal pyriform-shaped Purkinje cells with prominent nuclei (black arrow) and normal granular and molecular layers. (**E**) Semi-quantitative representation of degenerated Purkinje cells in different rat groups. Data are represented as mean ± SEM (*n* = 3); * *p* < 0.005 vs. control. H and E 200×, scale bar = 100 µm.

**Figure 16 biomolecules-09-00453-f016:**
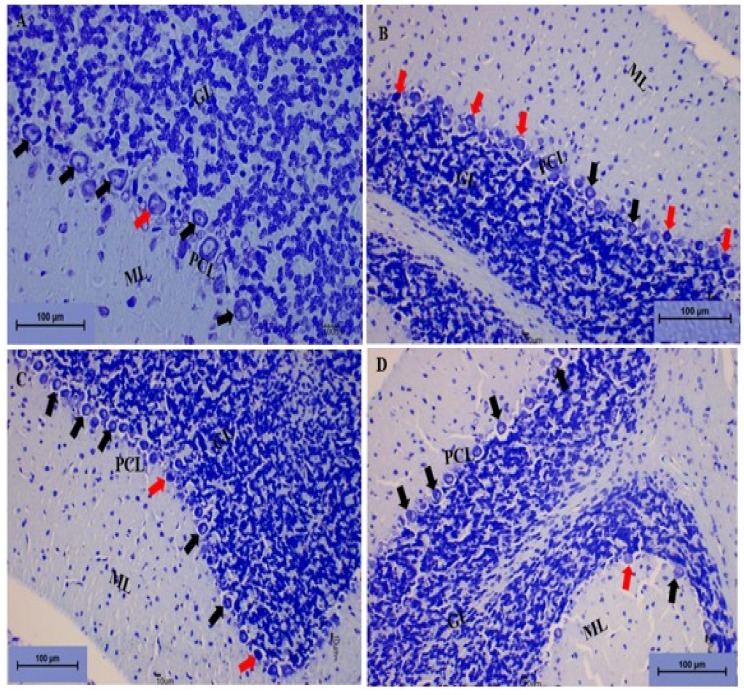
Photomicrograph sections of the cerebellums from the different rat groups. (**A**) Control group showing the white matter (W) and the three layers of the cerebellar cortex: The molecular layer (ML), the granular layer (GL), and the Purkinje cell layer (PCL) with the Purkinje cells (black arrow) showing prominent nuclei. (**B**) RC showing the Purkinje cells having deeply stained cytoplasm with distorted shapes (red arrow) and the molecular layer showing scattered glial cells (ML). (**C**) Cur100 and (**D**) Cur200 showing the restoration of the molecular layer (ML) and granular layer (GL); the Purkinje cell layer (PCL) appears to be arranged in a linear pattern with the Purkinje cells (black arrow) having prominent nuclei and a regular shape. Toluidine blue, 200×, scale bar = 100 µm.

**Figure 17 biomolecules-09-00453-f017:**
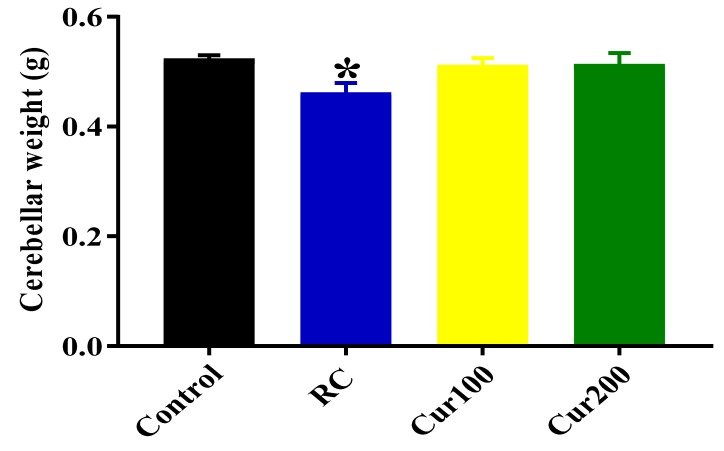
Effect of curcumin administration on cerebellar weights of Pb-induced rats. Data are represented as mean ± SEM (*n* = 6); * *p* < 0.05 vs. control.

**Figure 18 biomolecules-09-00453-f018:**
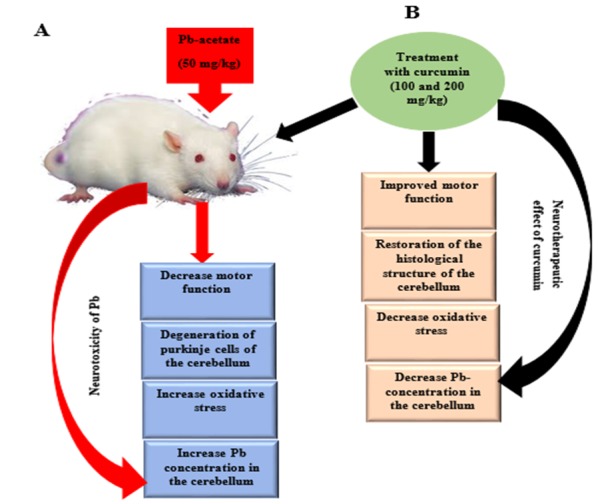
Proposed mechanism of Pb-induced cerebellar toxicity and the ameliorative effects of curcumin. Pb induction in rats, results in decreased motor function, degeneration of Purkinje cells of the cerebellum, increased oxidative stress, and increased Pb concentration in the cerebellum. Curcumin administration ameliorates the above-mentioned aberrations, which supports the neurotherapeutic effects of curcumin.

**Table 1 biomolecules-09-00453-t001:** Scoring procedures in the horizontal bar method.

Serial No.	Duration of Rat on Horizontal Bar	Score
1.	Falling time between 1 and 5 s	1
2.	Falling time between 6 and 10 s	2
3.	Falling time between 11 and 20 s	3
4.	Falling time between 21 and 30 s	4
5.	Falling time after 30 s	5
6.	The maximum score (5) was allotted if the rat placed one forepaw on the bar support without falling.	5

Note 1: If the experimental animal failed to grasp the bar properly and its was obviously due to error of the experimenter, the test was repeated after a brief rest by the rat to obtain a falling time greater than 5 s. Note 2: If the rat in three different attempts fell before 5 s and it appeared that it was not due to error of the experimenter in placing the rat on the bar, it was recorded and the best score was taken for the data.

**Table 2 biomolecules-09-00453-t002:** Work specification resume for ICP-MS Agilent 7700× and measurement parameters.

Parameters	Values	Units
Radio Frequency Power	1550	W
Radio Frequency Matching	1.6	V
Sample depth	9.5	mm
Torch	−0.1	mm
Torch-V	0	mm
Argon Gas Flow Rate	15	L/min
Carrier Gas Flow	1.01	L/min
Make up Gas Flow	0.15	L/min
Auxiliary Gas Flow Rate	0.58	L/min
Sample Uptake Rate	0.3	revolutions per seconds (rps)
Sample Uptake Rate	100	µL/min
Sampling Depth	6–7.6	mm
Spray chamber temperature	2	°C
Nebulizer Pump	0.1	revolution per seconds (rps)
Integration Time	3	s
Internal Standard (^103^Rh,^208^Bi)	200	part per billion (ppb)

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
