# Peer review of "Curcumin Attenuates Lead-Induced Cerebellar Toxicity in Rats via Chelating Activity and Inhibition of Oxidative Stress"

_biomolecules, 2019, doi:10.3390/biom9090453_

Round 1

Reviewer 1 Report

Submitted manuscript is considered as generally interesting and logical. Accordingly, this article would be acceptable after submit answers for following questions;

In the Materials and Methods, guideline of motor activity score should be described more precisely.

In Figure 4 and 12, authors should check SOD activity unit (µ/mg prot), such as µg or µmole?

In all bar graph figures, asterisk mark is difficult to find. Please, clear the asterisk symbols

In the histological Figure 7, 8, 15, 16, authors should submit more clear photomicrographs.

In the legends of Figure 11, 12, 13, 17, authors should change “pb induced” to “pb-induced”.

Authors assert that curcumin has the chelating activity via inhibitory action on Pb accumulation in cerebellar region by ICP-MS method. In this case, authors should submit direct proof on curcumin-Pb complex formation or citation references on the chelating action between curcumin and lead.

Reviewer 2 Report

Abubakar et al. investigated the effects of curcumin on lead-induced cerebellar toxicity in rats, assessing motor functions and suggesting chelating activity and inhibition of oxidative stress and chelating 3 activity as a possible mechanism of action.

The topic is a matter of interest because lead toxicity is a strongly significant environmental problem that can cause different problem both in central and peripheral nervous system.

Some data about protective effect of this compound on lead acetate-induced toxicity are already available (Sudjarwo et al. 2017, Benammi et al., 2017, Pal et al., 2015), but the authors present scientifically sound results and present new interesting data focusing on cerebellar toxicity and evaluating not only motor functions but even Pb concentration.

The manuscript is interesting and my only concern is about the title that is, in fact, ambiguous. The expression “via inhibition of oxidative stress and chelating activity” makes parallelism that implies curcumin acts via inhibition of chelating activity, that is far from correct.

The sentence should be reworded to make this clear. A possible solution could be “Curcumin attenuates lead-induced cerebellar toxicity in rats via chelating activity and inhibition of oxidative stress” but every restatement of the title to make the meaning clearer would be acceptable.

There are some minor mistakes or inconsistencies:

-         Line 28 and 96: should read thirty-six (hyphen missing)

-         Line 134: should read bicinchoninic acid assay

-         Line 233: missing space

-         Figure 15 E and Figure 18: please, improve definition.

Overall, I recommend publication with minor revisions.

Reviewer 3 Report

In this manuscript, authors examined the therapeutic effects of curcumin on Pb-induced cerebellar toxicity in rats. They found that curcumin treatment inhibits oxidative stress and reduces Pb concentration in the cerebellum thereby attenuating Pb-induced neurotoxicity and motor dysfunction. Although those results are interesting, there are several comments to improve the manuscript as follow;

1. There is no attempt to exam effects on the activation of astrocytes and microglia.

2. It is more informative to investigate if curcumin can facilitate the elimination of Pb in other tissues and blood.

3. Pb might damage cortex or hippocampus in addition to cerebellum. Does curcumin attenuating Pb-induced neurotoxicity in other brain region?

4. More detail method for quantification of the necrotic Purkinje cells should be described. Are they counted manually? Is it performed in a blinded manner?

5. It is unclear the age of animals used in this study. Do juvenile rats have severe damage by Pb-administration?

Reviewer 4 Report

In the manuscript entitled “Curcumin attenuates lead-induced cerebellar toxicity in rats via inhibition of oxidative stress and chelating activity” Abubakar and colleagues evaluated the therapeutic effect of curcumin on a rat model of Pb-induced neurotoxicity. The authors found that curcumin treatment improved the motor deficits induced by Pb poisoning. Moreover, curcumin reduced Pb concentration and neurotoxicity in the cerebellum.

I found the study well carried out and the topic interesting. However, some concerns need to be addressed before the manuscript acceptance.

Concerns

1.      I believe that the paper cited by the authors at page 2, line 63 (citation [8]) does not refer to African but African American children. Please check.

2.      Some sentences appear too long and not clear (see for example page 2, lines 66-70). Please check for spelling, grammatical and stylistic English mistakes.

3.      Although the authors cited proper papers on the doses of Pb acetate and curcumin used in the study (references 19-21), they should spend some more words about it. For example, is the administration of Pb acetate dose of 50 mg/kg an attempt to mimic Pb poisoning in humans? Please address this point.

4.      The quality of some figures appears poor.

For example, in figures 4, 5, 6, 9, 12, 13 and 17 the asterisks are barely visible, as well as the writing “GL” in figure 7A and 7B.

In the upper part of figure 7B, I believe that “GL” refers to “ML”.

Figure 15E looks blurry.

The picture depicted in figure 16A appears completely different from the pictures shown in 16B, C and D. Moreover, I cannot see the scale bar in the figure.

Round 2

Reviewer 3 Report

Authors have sufficiently revised the manuscript.